# Associations between the Willingness to Donate Samples to Biobanks and Selected Psychological Variables

**DOI:** 10.3390/ijerph19052552

**Published:** 2022-02-23

**Authors:** Jakub Pawlikowski, Michał Wiechetek, Anita Majchrowska

**Affiliations:** 1Department of Humanities and Social Medicine, Medical University of Lublin, 20-093 Lublin, Poland; anita.majchrowska@umlub.pl; 2Biobanking and Biomolecular Resources Research Infrastructure Poland, BBMRI.pl Consortium, 54-066 Wrocław, Poland; 3Institute of Psychology, The John Paul II Catholic University of Lublin, 20-950 Lublin, Poland; wiechetek@kul.pl

**Keywords:** biobank, willingness to donate, values, trust

## Abstract

Over the past few decades, there has been a dynamic development of biobanks collecting human biological material and data. Advances in biomedical research based on biobanks, however, are highly dependent on the successful enrolment and participation of human subjects. Therefore, it is crucial to recognise those factors affecting the willingness of individuals to participate in biomedical research. There are very few studies pointing to the role of trust, preferred values and specific psychological factors. The aim of our study was the analysis of the most significant relationships between selected moral and psychological variables (i.e., preferred values, types of trust and personality) and willingness to donate biological material to biobanks. The research was carried out on a Polish representative national sample of 1100 people over 18 years of age. Statistical methods with regression models were used during the analyses. The willingness to donate samples to a biobank was associated with different types of trust and specific values. Based on regression analysis, the most important factors related to the willingness to donate material to biobanks seemed to be (1) trust towards scientists and doctors and (2) selected preferred values such as knowledge, self-development and tradition. Other values or personality traits did not seem to be as important in this context. The obtained results can be useful in building the social responsibility of biobankers and scientists, issuing more appropriate opinions by research ethics committees and planning better communication strategies between participants and biobanks.

## 1. Introduction

Over the past few decades, there has been a dynamic development of biobanks collecting human biological material and data (population-based, disease-centric, genetic, etc.) that broaden knowledge about the genetic, behavioural and environmental determinants of many diseases; support the development of new biomarkers and drugs; validate laboratory and clinical findings; define new therapeutic targets; improve personalised medical care [1,2]. Biobanks are also useful for psychiatric and psychological analyses such as the role of personality traits in smoking [3], reporting adverse drug reactions [4], the polygenic risk for schizophrenia [5] and depression [6,7,8].

Advances in biomedical research based on biobanks are highly dependent on the successful enrolment and participation of human subjects. Therefore, it is crucial to recognise those factors influencing the willingness of individuals to participate in biomedical research. However, social considerations on biobanks are dominated by ethical and legal issues regarding the scope of informed consent, biosharing, the commercial use of samples and data, ownership issues, returning results, incidental findings, data management and privacy protection [9,10,11,12,13,14,15,16]. There are much fewer studies considering the societal and psychological factors influencing donation.

A few studies indicate that demographical variables, such as gender, education level and socioeconomic status, can play a role in willingness to donate. Those more favourable toward donation are middle-aged (usually 40–65 years old) persons [16,17] with higher education [18] and higher economic status, who live in urban areas and have children [19]. On the other hand, being a member of an ethnic minority is associated with lower willingness [20]. In our review of 61 previous social research studies, it was found that donation is a complex process that may be determined by psychosocial factors such as people’s knowledge and positive opinions of biobanks, trust, beliefs about the expected benefits, access to information about the research, donors’ cultural and religious beliefs and privacy protection [21,22,23,24,25,26,27]. A few studies reported donors’ motivations [27,28,29,30,31]. It is also known that knowledge about psychosocial attitudes toward biobanking may increase the effectiveness of the recruitment process [32]. The abovementioned studies, however, used simple methods for measuring psychosocial variables, mostly by single-question measurement. Only a few psychological reports were found that indicate the role of social trust, social involvement, the preferred value system and a pro-social attitude [18,33,34]. There is lack of psychological analyses based on a representative social sample considering personality, preferred values and specific types of trust that may relate to the donor’s behaviour.

From the psychological perspective, the donation of biological material to a biobank may be comprehensively analysed by referring to organ donation theories or to models explaining intentional behaviour, e.g., reasoned action approach [35,36,37,38]. According to these theories, a donation may be determined by several factors such as personality traits, behavioural traits, perceived norms, preferred values, beliefs about control, behavioural beliefs (benefits and risks), normative beliefs, cultural beliefs and search for meaning.

The aim of our study was the analysis of the relationship between selected psychological variables and the willingness to donate biological material to biobanks in the Polish population. It was hypothesised that the willingness to donation is associated with some types of trust, specific personality traits, approach to meaning in life and one’s preferred value system. The research was conducted as part of a task dedicated to the ethical, legal and societal implications of biobanking within the remits of a project of establishing the Biobanking and BioMolecular Resources Research Infrastructure–European Research Infrastructure Consortium (BBMRI-ERIC) [39].

## 2. Materials and Methods

This research was carried out on a group of 1100 people over 18 years of age (Table 1). The sample was random, and the selection of respondents corresponded to a representative structure of the Polish population in the following areas: sex (100% compliance the local data bank (LDB)), age (maximum deviation 2% from LDB), number of respondents in a given voivodship calculated on the basis of the population distribution throughout the country (100% compliance with the LDB), place of residence (maximum deviation 1% from the LDB) and level of education (maximum deviation 3% from the LDB). The sample was primarily selected using random-route as a default method (employing the computer-aided personal interview (CAPI) technique). Beginning at the starting point (the first house number on the selected street), the interviewer visited every third residential premise (flat/detached house) until collecting a maximum of three respondents on the given street or exhausting the pool of addresses where respondents fulfilling the study’s inclusion criteria could stay. The maximum number of people from one locality amounted to nine respondents in cities/towns/villages with up to 100,000 people and fifteen respondents in cities/towns with over 100,000 inhabitants. The response rate was 72%. The maximum acceptable statistical error of the measurement was 4% with a confidence interval of 95%. The questionnaire consisted of several parts relating to the willingness to donate samples to a biobank and psychosocial variables. Participation in the study was voluntary.

### 2.1. Measures

Willingness to donate biological material to a biobank was measured using the following question: “Please imagine that a biobank from the nearest provincial city, operating at a medical university, asks you to donate a blood sample for research. Approximately 30 mL of blood (three large tablespoons) will be drawn, and an interview will take place regarding health- and disease-related issues such as lifestyle (e.g., eating habits, exercise, use of stimulants and sleep), environment, drug use and medical history. The collected samples and data will then be made available to scientists for research in an anonymised form (i.e., the donor cannot be identified). Would you give a blood sample to a biobank in the situation described above?” The respondents were asked to rate their willingness to donate on a five-point scale from 1 (definitely not) to 5 (definitely yes).

Personality traits were measured using the TIPI questionnaire [40]. This consists of ten items that are rated on a scale from 1 (strongly disagree) to 7 (strongly agree). The method is based on the concept of the Big Five and assesses five personality dimensions: emotional stability, extraversion, agreeableness, openness to experience and conscientiousness.

Trust was measured in three aspects: trust in doctors, trust in scientists and trust in other people. The respondents were asked to answer three questions (“*Please rate your trust in….”*) relating to a specific group on a scale from 0 (“I do not trust at all”) to 10 (“I trust completely”).

The tendency towards risk was measured with a single question (“*Please rate your risk tendency…”*) rated on a seven-point scale from 1 (“I am a risk-averse person”) to 7 (“I am a risk-seeker person”).

Meaning in life was measured using two items inspired by the Meaning in Life Questionnaire that measures two aspects of meaning in life [41]. The presence of the meaning in life was measured by the question: “*To what extent does the phrase: ‘I am satisfied with life*’ *match with your approach*?”. The search for meaning in life was measured by the question “*To what extent does the phrase: ‘I am seeking for a purpose or mission in life*’ *match with your approach*?”. The respondents assessed these questions on a seven-point scale from 1 (definitely not) to 7 (definitely yes).

The hierarchy of preferred values was measured with a list of 17 different values (i.e., safety, life, security, modernity, state, money, travels, helping others, work, nature, religion, family, personal development, respect of others, art, tradition, knowledge and health) taken from public surveys carried out in Poland. The respondents were asked to respond to the question: “*To what extent are the following values important for you in your life?*” The answers were given on a five-point scale from 1 (definitely not important) to 5 (definitely important).

The analysis covered sociodemographic data, such as gender, age, education, place of residence, self-assessment of material conditions (scale from 1 (very bad) to 6 (very good)) and self-assessment of health (scale from 1 (very bad) to 6 (very good)).

### 2.2. Statistical Analysis

Data were analysed using the IBM SPSS Statistics v.25 [42]. Descriptive statistics (frequency, mean, percentage and standard deviation), the Student’s *t*-significance test for independent samples, r-Pearson, rho-Spearman correlation coefficients and the regression with stepwise input method were used during the analyses. The level of statistical significance was *p* < 0.05.

## 3. Results

The analysis of the results was divided into four parts. In the first part, the level of willingness to donate samples to a biobank was analysed. In the second, the relationship between willingness and socio-demographic variables was assessed. The third describes the link between willingness and psychological characteristics. The final part indicates the psychological variables that best explain the willingness to donate biological material to biobanks.

The willingness to donate samples to a biobank varied. Approximately half of the respondents declared that they were open to donating (“rather yes”—30.4%; “definitely yes”—17.2%). Persons unwilling to donate a sample constituted approximately 28% of the respondents (“definitely not”—3.2%; “rather not”—24.9%). Approximately a quarter of the surveyed people did not have a clear opinion (“difficult to say”—24.4%).

No statistically significant relationships were found between the willingness to donate samples to a biobank and sociodemographic variables, i.e., gender (*t* = 0.734, *p* = 0.486), age (*r* = 0.052, *p* = 0.087), place of residence (*rho* = 0.042, *p* = 0.166), education (*rho* = 0.001, *p* = 0.994), self-assessed material condition (*r* = 0.020, *p* = 0.500) and self-assessed health (*r* = −0.034, *p* = 0.264).

Selected psychological variables turned out to be factors significantly related to the willingness to donate biological material to a biobank (Table 2). The willingness to donate samples to a biobank was significantly positively associated with all types of trust measured (in doctors, scientists and other people). The strongest relationship was observed for trust in doctors and scientists. Another factor, positively related to the willingness to donate a sample, was the tendency towards risk. Regarding the preferred values, a significant positive correlation was observed between the willingness to donate and such values as: work, helping others, travels, personal development, tradition and knowledge (but not health or life). No statistically significant relationships were found between the willingness to donate samples and personality traits based on the Big Five model.

A stepwise regression analysis was performed to extract those variables that best determined the willingness to donate samples to a biobank. The explained variable was the willingness to donate a sample and the explanatory variables were types of trust, tendency towards risk, personality traits, sense of and searching for meaning in life and preferred values. A statistically significant model was obtained (*F* = 10.70, *p* = 0.010; *R^2^* = 0.034) in which the willingness to donate biological material to a biobank was best predicted by trust in scientists (*β* = 0.082, *t* = 2.134, *p* = 0.033) and doctors (*β* = 0.080, *t* = 2.089, *p* = 0.037) as well as preferred values such as personal development (*β* = 0.086, *t* = 2.854, *p* = 0.004) and tradition (*β* = 0.060, *t* = 1.982, *p* = 0.048).

## 4. Discussion

Participation in different types of biomedical research is considered one of the main challenges facing researchers in medicine and epidemiological fields [43]. Inadequate involvement in biomedical research may affect the power of a study, increasing the likelihood of type II errors, and adversely influencing the generalisability of results to the general population [44]. Therefore, it is important to know those significant factors that can improve participation.

The conducted research confirmed the relationship between the willingness to donate biological material to a biobank and trust, selected preferred values and other specific personal features. Earlier research has also suggested a relationship between donation and trust in the context of biobanking [18,30,31,45]. Our research, however, revealed that not only general trust in people was important but specific kinds of trust, such as trust in scientists and doctors, play a more important role. This relationship may suggest that individuals open to participating in biobanks may be particularly interested in the development of science and medicine. This interpretation was also confirmed by the observed significant link with the high preference for the value of knowledge and personal development. At the same time, trust may also be associated with the sense of security provided to individuals by professionals (scientists and doctors) working in a biobank. Therefore, relying on the authority of scientists and doctors may be the key to success in conducting awareness-raising social campaigns about biobanks and in encouraging potential participants to donate. An example of effective cooperation with participants is provided by an Estonian population-based biobank, where a significant number of donors were enrolled through contact with general practitioners (GPs) [46].

Biobankers, however, should be aware that people trust public institutions more than commercial and foreign institutions [21,28,47,48,49,50]. Lower trust has been observed among ethnic minorities (i.e., African Americans, Mexican Americans, Native Americans, and Hawaiian and Alaskan Natives), which may be a consequence of their negative experiences with colonisation, eugenics and medical experiments. Therefore, a very special role in participants’ enrolment can be played by researchers and physicians who have authority in a given country or minority, or even who belong to these communities. Trust in biobank research can diminish when the use of material is incompatible with participants’ expectations [31].

Our study showed that openness to donating samples to a biobank was significantly related to such values as work, personal development, helping others, knowledge and travel as well as tradition. In the case of the first four categories, it may be related to a general openness to people and the world, the progress of science and development of medicine. Several other studies have confirmed that many donors are motivated by altruistic premises [27,28,29], helping others [51], a general feeling of duty [52] and the desire to contribute to new knowledge, new treatments and the common good [24,26]. An association between willingness to donate and prosocial values has also been observed [28]. Several people, however, also expect benefits for their families, relatives and ethnic groups, or their desired medical services and research results [21]. The preference for the value of travel in the group of potential donors is also linked with a tendency towards risk. Therefore, the reference to the preferred values of potential donors in social is worth considering campaigns with appropriate metaphors (e.g., adventure, journey, development, helping others, and expanding knowledge).

The interpretation of the link between the values of tradition and willingness to donate was less obvious. Perhaps, this may be the consequence of the association between tradition and the solidarity of the group which results in taking actions aimed at group survival [53]. Such a relationship (between openness to biobanking and tradition) may also result from the perception of biobanks as offering the option of performing genetic research, especially genealogical analyses. Other studies suggest that donors were interested in genetic research results [21,54], and some biobanks offer genealogical analyses based on the collected samples [55]. Therefore, it seems that the offer of performing genealogical tests in biobanks may be one of the important factors that encourage donation.

A relationship between the willingness to donate material to a biobank and the tendency towards risk was observed. This is consistent with the results of other studies [18,25,56]. It may be hypothesised that biobanks are still perceived as a novelty or a mystery [25,50], and enrolment is associated with some risks, but it is also a form of adventure and participation in scientific discoveries in medicine. Such an interpretation was also confirmed by the observed positive relationship with the value of traveling and experiencing something new. Many donors are aware that there are risks associated with biobank participation associated not only with sampling but also with data management such as privacy breaches. They are, however, open to collaboration [18,21,32,57].

It may be surprising that there was no relationship observed with the values of health and life. Perhaps, in public opinion, biobanking is viewed as a strictly scientific rather than a medical matter. We also did not observe a relationship between willingness to donate and the value of religion. Such relationships were observed in some studies [19,58]; however, in others, religious beliefs did not seem to influence donors’ decisions [59,60] and, even fewer religious persons were more interested in donation [25,30]. This may be the result of social and cultural differences associated with different types of spirituality or religiosity, e.g., in Malaysia differences in attitudes towards biobanking between Christians and Hindus were observed [23]. The lack of this relationship in Poland may also be the effect of the little religious teaching activity of the Catholic Church (as the main religious denomination in Polish society) in the field of scientific research. For this reason, the perception of ethical problems in the area of biobanking is not shaped by religious premises. In the public view, probably, the entire question of HBM biobanking is perceived more as a scientific issue rather than an ideological one.

In our research, the willingness to donate material to a biobank was not associated with the sense of meaning in life. A weak relationship, however, was observed with the search for meaning in life. This means that some people may perceive their participation in biobanking as an opportunity to find an important goal or kind of personal mission focused on scientific development and helping others. Such an interpretation may be consistent with the theoretical assumptions related to the understanding of the search for meaning in life as the effort to establish or expand the meaning and purpose of one’s life [61].

There was no significant relationship between personality traits and the willingness to donate samples to a biobank in our study. This may be surprising since some personality traits (e.g., agreeableness) were linked with pro-sociality in another study [62]. This lack of relationship may suggest that personality is not a direct predictor of enrolment in biobanking and that other moderating factors should be investigated. This would be consistent with the assumptions of the reasoned action approach.

The regression analysis indicated that the willingness to donate material to a biobank is best explained by trust in scientists and doctors, and preferred values such as personal development and tradition (among other analysed variables). This means that such factors can play an important role in enrolling potential participants in biobanks. The level of explanation, however, is not very high, which may indicate that the act of donation is the result of many factors that should be the subject of further research using the moderation approach, including more variables as moderators, such as type of donated tissue, the purpose of the research, place of research, feeling of security, access to information about research results and others [21,24]. For instance, people are more willing to donate blood, cancer tissues, saliva, urine, skin and kidney tissues but are less willing to donate bones, organs from the deceased, or germ cells left over from in vitro [25,26,27]. Many declare donations for cancer research, while stem cell research, cloning, genetic engineering, research involving the combination of human samples with animals, or even research conducted abroad are much more controversial [24,25,63]. Most respondents also objected to research with stigmatising potential, i.e., on mental disorders, intelligence, homosexuality, or which had commercial implications [32]. Further, donors may be discouraged by their disapproval of the purpose of the research, concerns over the safety of the data [28,30], fear over the invasive nature of the sampling procedure (pain, sight of blood, needle injections) [29], fear over infection with HIV, the detection of genetic predispositions, the use of the sample contrary to the donor’s values [22], commercial use of their samples and geographical distance from the biobank [57]. Willingness to donate samples correlated with better knowledge and positive opinions on biobanks [64]. In a pan-European study, only 10% of respondents who had never heard of biobanks would not donate [18]. Over time, it is possible to specify more patients’ preferences and priorities in research based on biobanking and build more trust in communication with participants and society [65].

In European countries we observe societal variation in the will to donate samples to a biobank. In Scandinavian countries, 83% of Finns and 86% of Swedes declared willingness to donate [50,66], while only 4% of Greeks did [18]. In the UK, almost 75% of respondents agreed with donation, while 18% did not [67]. In Latvia the number of respondents who would be willing to collaborate with biobanks increased from 25.8% to 40.7% since 2010 to 2019 [68]. The willingness to donate in the Polish population was lower than in Scandinavian countries, but it seems to be higher than in southern Europe and other Central European countries.

## 5. Conclusions

The presented study regarding the willingness to donate samples in biobanking in the context of psychological variables was the first study on this topic in Europe that was conducted on a representative national sample. The study included several psychological variables that have so far been scarcely analysed in the context of biobanking. The meaning of the term “biobank” was mapped, which gave the respondents an opportunity to refer to the defined research subject.

Among the limitations of our study was the fact that it was conducted on the general population and not on actual donors. This will be a task for future research, as biobanks better develop in Poland. The method (personal interviews) could also have influenced the results. Another weakness may be the method used to measure personality. It is often used in polls, but its psychometric parameters are low, which could have resulted in the lack of significant relationships between personality traits and the willingness. Few variables were measured by one item (due to the PAPI method), which could have affected the reliability of the measurement. In the future, tools with a larger number of items should be used. The obtained results may not be the same in other populations due to moderation by other cultural factors.

The willingness to donate biological material to a biobank may be determined by many psychological factors. One of the most important factors related to the willingness to donate material to biobanks seems to be trust towards scientists and doctors. Selected psychological traits, such as preferred values (knowledge, self-development and tradition) may significantly shape the willingness to donate material to a biobank. Other values or personality traits did not seem to be as important in this context. The lack of a statistically significant correlation between the willingness to donate and personality may be the result of measurement method which was used. TIPI is a simple tool that measures each personality dimension with only two items, so its reliability is quite low. This may also be due the personal developmental period. At different stages of life, personality traits and preferred values may influence decisions differently. In future research, it would be worth examining the donation in a more precise way and asking about the specific type of human biological material.

The obtained results can be useful in building the social responsibility of biobankers and scientists and planning communication strategies between biobanks and potential participants, based on the authority of researchers and doctors as well as the values preferred in this group. It would be worth using such authorities in social campaigns and the spots in media. There should be a possibility of face-to-face meetings between potential participants and researchers. It also seems important to show the specific benefits that can flow among people, society, and next generations from biobanks.

## Figures and Tables

**Table 1 ijerph-19-02552-t001:** Sample characteristics (*N* = 1100).

Variables		*n* (%)/M (SD)
Age		47.41 (17.37)
Gender:	Women	575 (52.3%)
	Men	525 (47.7%)
Education:	Primary or vocational	79 (7.2%)
	Secondary	596 (54.2%)
	High	425 (38.6%)
Place of residence:	Village	158 (14.4%)
	City up to 50,000 residents	249 (22.6%)
	City from 50,000 to 100,000 residents	169 (15.4%)
	City with over 100,000 residents	524 (47.6%)
Self-assessment of material conditions:	Very bad	21 (1.9%)
	Bad	28 (2.5%)
	Rather bad	183 (16.6%)
	Rather good	602 (54.7%)
	Good	228 (20.7%)
	Very good	38 (3.5%)
Self-assessment of health:	Very bad	20 (1.8%)
	Bad	38 (3.5%)
	Rather bad	155 (14.1%)
	Rather good	531 (48.3%)
	Good	283 (25.7%)
	Very good	73 (6.6%)

**Table 2 ijerph-19-02552-t002:** Willingness to donate a sample to a biobank and selected psychological variables (Pearson’s *r* correlation coefficient).

Variables	Descriptive Statistics		Willingness to Donate	
	M	SD	*r*	*p*
Willingness to donate	3.33	1.12		
Personality traits:				
Emotional stability	4.29	0.77	−0.02	0.52
Extroversion	4.28	0.74	0.01	0.76
Agreeableness	4.66	0.69	0.01	0.84
Openness to experience	4.49	0.73	0.04	0.22
Conscientiousness	4.77	0.75	0.05	0.11
Tendency towards risk	3.53	1.61	0.09 **	0.00
Types of trust:				
Trust in other people	5.55	2.31	0.08 **	0.01
Trust in doctors	6.04	2.33	0.14 **	0.00
Trust in scientists	6.52	2.30	0.14 **	0.00
Types of meaning in life				
Meaning in life/satisfaction with life	3.69	0.88	0.03	0.28
Searching for meaning in life	3.40	1.02	0.06 *	0.05
Preferred values:				
Security	4.66	0.69	0.06	0.05
Modernity	3.91	0.87	0.02	0.51
State	3.87	0.97	0.04	0.18
Money	3.96	0.79	0.03	0.27
Travels	3.62	1.01	0.06 *	0.04
Helping others	3.82	0.88	0.06 *	0.04
Work	3.83	1.02	0.06 *	0.04
Nature	4.22	0.81	0.05	0.13
Religion	3.02	1.40	−0.02	0.44
Family	4.59	0.70	0.01	0.64
Personal development	4.22	0.82	0.11 **	0.00
Respect of others	4.22	0.83	0.04	0.19
Art	3.56	0.95	0.00	0.91
Tradition	3.68	1.02	0.09 **	0.00
Knowledge	4.40	0.73	0.07 *	0.02
Health	4.71	0.65	0.06	0.06
Life	4.69	0.65	0.03	0.25

* *p* < 0.05; ** *p* < 0.01.

## Data Availability

All data generated or analysed during this study are included in this published article.

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
