# Peer review of "Associations between the Willingness to Donate Samples to Biobanks and Selected Psychological Variables"

_ijerph, 2022, doi:10.3390/ijerph19052552_

Round 1
Reviewer 1 Report
Dear authors,
I would suggest being more precise in explaining this claim: "Other values, or personality traits based on the Big Five concept, do not seem to be so 21
important in this context." Why is the Big Five model not relevant for your study? The Big Five concept is mentioned three times in the article, and I think the paper would benefit from a deeper conceptualization of why the model is not relevant for your results.
For future studies, it could be beneficial to consider including other genders other than "men" and "women", given that other options ("non-binary", "gender non-conforming", "gender diverse") are being increasingly chosen by people and even a legal option in some European countries. This would add another layer of complexity to research, and would allow getting data on underrepresented groups.
Best regards,
Author Response
Dear authors,
I would suggest being more precise in explaining this claim: "Other values, or personality traits based on the Big Five concept, do not seem to be so important in this context." Why is the Big Five model not relevant for your study? The Big Five concept is mentioned three times in the article, and I think the paper would benefit from a deeper conceptualization of why the model is not relevant for your results.
For future studies, it could be beneficial to consider including other genders other than "men" and "women", given that other options ("non-binary", "gender non-conforming", "gender diverse") are being increasingly chosen by people and even a legal option in some European countries. This would add another layer of complexity to research and would allow getting data on underrepresented groups.
Answer:
Thank you very much for your review and remarks. We agree that the association between the willingness to donate and personality traits seems a more complex problem (like values). Personality traits may influence many areas of human behaviour, but in the context of biobanking it may not work in a direct way - we have added this information in the text. We have also referred to this issue in conclusion where we have added that: “The lack of a statistically significant correlation between the willingness to donate and personality may be the result of measurement method which was used. TIPI is a simple tool that measures each personality dimension with only two items, so its reliability is quite low. This may also be due the personal developmental period. At different stages of life personality traits and preferred values may influence decisions differently”.
Regarding the gender variable in our study, we would like to emphasize that we did not exclude from the study anyone who would declare a different gender other than a woman or man. This variable also did not play a significant role in shaping attitudes towards biobank in other studies (which was confirmed in ours). We will remember, however, about diversity of gender in future studies.
Reviewer 2 Report
The manuscript by Pawlikowski et al is aimed to evaluate a very important factor in the development of Biobanks, the willingness of donor to give their biological samples for research. The identification of the barriers for donation could help to design strategies to overcome them.
Major revisions:
The authors should clarify in depth the statistical analysis they have carried out to make the correlation between psychological barriers and willingness to donate. There are a lot of correlations described with a p value < 0.05 but the strength of the correlation coefficient is very weak. In that line, they should also modify the table 2 since it is not possible to follow the important results in the present design. The Legend of the Table must be modified, and a clear explanation of the variables included should be added.
The result section should be improved with a more detailed description of the findings obtained.
In the discussion the authors should emphasized the findings of the present study and if there is some perspective regarding the possibility to modify the barriers to donate biological samples to biobanks.
Minor revision:
As a supplementary material it could be helpful to include the questionnaire used in the study.
Author Response
Major revisions:
The authors should clarify in depth the statistical analysis they have carried out to make the correlation between psychological barriers and willingness to donate. There are a lot of correlations described with a p value < 0.05 but the strength of the correlation coefficient is very weak. In that line, they should also modify the table 2 since it is not possible to follow the important results in the present design. The Legend of the Table must be modified, and a clear explanation of the variables included should be added.
The result section should be improved with a more detailed description of the findings obtained.
In the discussion the authors should emphasized the findings of the present study and if there is some perspective regarding the possibility to modify the barriers to donate biological samples to biobanks.
Answer:
Thank you very much for your review and remarks. Linguistic errors have been corrected and the article has been revised by an English native speaker. The table was modified to improve its readability - unnecessary information has been deleted and the legend was improved. We agree that the strength of the correlation coefficient is very weak. In our opinion it may be the effect of the research sample which was national representative, but very diverse some variables (e.g. age) which could reduce the value of correlation coefficients (e.g. different values may be preferred by young people than by elders). We suggest in the manuscript that further research should take into account moderating variables - they may allow to obtain more significant correlations in specific groups and for specific kind of biological material. In the discussion we emphasized that, based on the research results, it will be possible to reduce barriers and improve communication with potential donors, based on the authorities they trust the most. The questionnaire was in Polish, therefore we are not convinced that its enclosure as additional material would have any significant value for English readers. The analyzed variables were described in detail in the methodological and result sections.
Reviewer 3 Report
Thank you very much for your efforts; this paper needs more attention.
it is written in very bad English and one has to suffer to understand what the authors are trying to say. The topic of the article is very interesting but it has to be presented in good English the least.
Author Response
Thank you very much for your efforts; this paper needs more attention.
it is written in very bad English and one has to suffer to understand what the authors are trying to say. The topic of the article is very interesting but it has to be presented in good English the least.
Answer:
Thank you very much for your review. We improved the quality of language. Linguistic errors have been corrected and the article has been revised by an English native speaker.
Reviewer 4 Report
Manuscript describes actual problem, several items should be explaned or corrected. why do you scale some questions 1-5, 1-6,...1-10 it makes the evaluation more difficult. How was the studied cohort obtained, what were the criteria despite of the "random nature".
In discussion and conclusions some items such as "trust in doctors" , "trust in scientists" are explaned in great details, some are almost neglected, e.g. money, work, nature, family,...Especially town versus village, age and religiosity are important to explane.
As regards data associated with samples, it should be added and explaned, they have great influence on the willingness to donor samples.
Only a few words in devoting comparing with similar European studies, Malaisia, and Africa are not well comparable.
Future perspectives should be given, based on current results.
Author Response
Comments and Suggestions for Authors
Manuscript describes actual problem, several items should be explaned or corrected. why do you scale some questions 1-5, 1-6,...1-10 it makes the evaluation more difficult. How was the studied cohort obtained, what were the criteria despite of the "random nature".
In discussion and conclusions some items such as "trust in doctors" , "trust in scientists" are explaned in great details, some are almost neglected, e.g. money, work, nature, family,...Especially town versus village, age and religiosity are important to explane.
As regards data associated with samples, it should be added and explaned, they have great influence on the willingness to donor samples.
Only a few words in devoting comparing with similar European studies, Malaisia, and Africa are not well comparable.
Future perspectives should be given, based on current results.
Answear:
Thank you very much for your review and suggestion. Different response scales were used in the study due to the fact that several measures (e.g. TIPI scale) were taken from the original version of the method and some variables were used by us in other studies and we wanted to compare their results. We assume that the different scales for questions had no influence on the result of the study, as the questionnaire contained detailed instructions. The selection of the sample was based on the criteria of representativeness for Polish population according to: age, gender, education and place of residence. We added more clarifying information in method section.
In discussion and conclusions we focused on the most significant variables associated with willingness to donate. We discussed religiosity, however in our study we did not observe such significant relationship with donation as in other studies (we hypothesise that it is associated with the lack of official statement of Catholic Church in this issue). The lack of a clear relationship between donation and socio-demographic variables, e.g. place of residence, is probably related to the specificity of the situation in Poland (we observe a migration from cities to the countryside and people living in the countryside are very diverse not only in economic terms, but also in terms of education). We have mentioned in introduction and discussion about data associated with samples as an important barrier for potential donors. Results from other European countries were described in discussion for a better comparison. We have added future perspective of research in conclusion and possible uses of the research results.
Round 2
Reviewer 2 Report
The authors have responded appropriately to all my comments, and have strengthened the manuscript.
Reviewer 3 Report
This is a poineering work on the willingess/ rejection to donate samples to biobanks in Europe. The author/s examine the reasons which encourge people (or desuade them from) to donate samples to biobanks. The final version of the paper, in its revised English, is much better than earllier versions. Now, readers can follow the line of thought in the paper and would be able to benefit from its findings.
(please check "in" line 97; I think it is "on".